# Tolerogenic Therapies in Cardiac Transplantation

**DOI:** 10.3390/ijms26093968

**Published:** 2025-04-23

**Authors:** Laurenz Wolner, Johan William-Olsson, Bruno K. Podesser, Andreas Zuckermann, Nina Pilat

**Affiliations:** 1Center for Biomedical Research and Translational Surgery, Medical University of Vienna, 1090 Vienna, Austria; 2Department of Cardiac and Thoracic Aortic Surgery, Medical University of Vienna, 1090 Vienna, Austria

**Keywords:** heart transplantation, immune tolerance, tolerogenic therapies

## Abstract

Heart transplantation remains the gold-standard treatment for end-stage heart failure, yet long-term graft survival is hindered by chronic rejection and the morbidity and mortality caused by lifelong immunosuppression. While advances in medical and device-based therapies have reduced the overall need for transplantation, patients who ultimately require a transplant often present with more advanced disease and comorbidities. Recent advances in tolerance-inducing strategies offer promising avenues to improve allograft acceptance, while minimizing immunosuppressive toxicity. This review explores novel approaches aiming to achieve long-term immunological tolerance, including co-stimulation blockade, mixed chimerism, regulatory T-cell (Treg) therapies, thymic transplantation, and double-organ transplantation. These strategies seek to promote donor-specific unresponsiveness and mitigate chronic rejection. Additionally, expanding the donor pool remains a critical challenge in addressing organ shortages. Innovations such as ABO-incompatible heart transplantation are revolutionizing the field by increasing donor availability and accessibility. In this article, we discuss the mechanistic basis, clinical advancements, and challenges of these approaches, highlighting their potential to transform the future of heart transplantation with emphasis on clinical translation.

## 1. Introduction

Cardiac transplantation remains the gold-standard treatment for end-stage heart failure; however, achieving long-term graft survival and preventing chronic rejection are formidable challenges. Standard immunosuppressive regimens, while effective in reducing early rejection, lead to generalized immune suppression, increasing the risk of infections, malignancies, and other complications. Moreover, these regimens often fail to prevent chronic rejection, resulting in progressive graft loss and limited patient outcomes over time. As a result, there is a critical need for therapies that can selectively induce immune tolerance to the transplanted heart to promote sustained graft acceptance, while preserving overall immune function.

The primary aim in preventing graft rejection is the induction of graft-specific immunological tolerance. While current immunosuppressive regimens significantly increase short term graft survival, they have major limitations like increased risk of infection and higher rates of malignancies. Additionally, while acute organ rejection can be prevented relatively well, immunosuppressive therapy does not impede chronic rejection and late graft loss, which remains a serious clinical problem [1].

Tolerance approaches in (cardiac) transplantation seek to re-educate the recipient’s immune system to not recognizing the allograft as a “threat”. Resulting non-responsiveness towards donor antigens would eliminate the need for continuous immunosuppression and reduce the risk of antibody mediated rejection (ABMR) [2]. Several strategies have emerged to achieve this, including co-stimulatory blockade, mixed chimerism, and the therapeutic use of T regulatory cells (Tregs).

A co-stimulatory blockade aims to prevent T-cell activation through the inhibition of key signaling pathways involved in immune response, notably the CD28/B7 and CD40/CD154 axes. This method has shown potential in prolonging allograft survival and preventing/delaying rejection when combined with other therapies [3,4]. Mixed chimerism, achieved through donor bone marrow (BM) transplantation after appropriate conditioning regimens, offers a robust approach to immune tolerance by educating newly emerging T-cells in the thymus to recognize the allograft as “self” [5]. Tregs are instrumental in maintaining immune homeostasis by suppressing effector T-cell responses and limiting inflammatory pathways, with studies demonstrating that Treg infusion and expansion can improve graft tolerance [6,7].

Despite advances in all those fields, the application of tolerogenic therapies in cardiac transplantation has lagged behind other organs, such as kidney and liver, due to the heart’s unique immunogenic profile and its resistance to tolerance [8]. In preclinical rodent models, the induction of tolerance towards cardiac allografts has proven particularly challenging, as the heart, in contrast kidney and liver, is not accepted spontaneously across major histocompatibility complex (MHC) barriers and is, therefore, highly susceptible to alloimmune responses [9]. This necessitates an exploration of novel tolerogenic pathways and combination therapies to address these barriers. In the clinical setting, spontaneous operational tolerance seems to be restricted to the liver [10]; however, many tolerance trials have been conducted in kidney transplantation, as dialysis is available as a rescue therapy.

This review will explore the current landscape of tolerogenic therapies in solid organ transplantation (SOT), especially the heart, examining new mechanistic insights into how co-stimulatory blockades, chimerism approaches, and Tregs could promote tolerance. Additionally, it will address recent advances in cellular and molecular therapies, possible ways of increasing the donor pool, and potential future directions for achieving durable graft acceptance in heart transplant recipients.

## 2. Co-Stimulation Blockade

It is well-established that T-cell activation and the initiation of an antigen-specific immune response require two critical signals. The first signal involves the T-cell receptor (TCR) interacting with either donor MHC molecules displayed on the graft or MHC molecules presented by the recipient’s antigen-presenting cells (APCs). The second signal, provided by co-stimulatory molecules, such as CD28 and its ligands CD80 and CD86, on APCs, as well as CD40 and its ligand CD154, drives the antigen non-specific response. Blocking this secondary signal through a co-stimulation blockade suppresses the T-cell response, which initially sparked significant interest among researchers and clinicians as a way toward achieving long-term tolerance and preventing rejection (selected biologicals blocking co-stimulation are shown in Figure 1).

While simultaneous anti-CD154 antibody and anti-CD28 treatment could induce tolerance towards skin allografts in mice and could prevent chronic vasculopathy in murine cardiac allografts, it fell short of inducing tolerance in primates, even though it extended heart allograft survival significantly [11,12]. In a study using a combination of CD154 antibodies with CD122, CD252, and LFA-1 blockades, murine heart transplant survival was extended to an average of 22 days without additional immunosuppression [13]; however, high antibody levels persisted, and all grafts were eventually rejected. These findings underscore that co-stimulation blockade alone was insufficient for durable graft acceptance, as cessation of the treatment consistently led to rejection. One major challenge is the accumulation of memory T-cells, which are less dependent on co-stimulatory signals for activation [14]. Further, human platelets express CD154, which hindered efforts to translate anti-CD154 into the clinic due to increased thromboembolic events [15]. By now, so called second-generation anti-CD40L antibodies, TNX-1500, Tegoprubart, and Dazodalibep, are being investigated (selected clinical trials are listed in Table 1).

They have modified Fc portions and, hence, are not thrombogenic. TNX-1500 was tested in monkey heart allograft recipients were the hearts that showed good function, and no adverse effects were observed [16]. TNX-1500 monotherapy could prevent the production of DSA and inhibited chronic rejection, paving the way for the first human trials [16]. Tegoprubart and Dazodalibep are currently being tested in phase II trials for patients after kidney transplantation [17].

Another option to inhibit CD40/CD40L signaling is by blocking CD40. CD40 is not expressed on platelets, and these antibodies thereby do not cause thrombotic events. However, the transplantation of murine CD40−/− bone marrow into wildtype recipients did not induce chimerism and tolerance [18]. The engraftment of allogeneic BM was only possible when both the direct and indirect pathway of the CD40 signal was blocked, meaning that the donor and recipient were lacking CD40.

Regarding CD40 antagonists, to date, Bleselumab and Iscalimab have been extensively studied in SOT, while KPL-404 has been investigated in only a single xenotransplantation study [17]. Iscalimab is inferior compared to tacrolimus-based therapy in preventing acute kidney graft rejection, and Bleselumab has shown inconsistent results in different clinical trials [17].

KPL-404 was used in combination with MMF and corticosteroids in the first pig-to-human cardiac xenotransplantation, where graft function was preserved for up to seven weeks until ABMR, and a viral infection resulted in graft failure [19].

The OX40–OX40L pathway functions as a secondary mechanism for (memory) T-cell activation, following the co-stimulatory signaling of traditional pathways. Antibodies against OX40L inhibited the activation of memory T-cells and showed in a murine skin transplant model, in combination with co-stimulatory blockade, prolonged graft survival and less donor-reactive T-cells [20,21]. Moreover, in a murine heart transplant model, anti-OX40L mAb in combination with ethylene carbodiimide-fixed donor splenocytes promoted donor-specific tolerance in pre-sensitized mice [22]. Another experiment proved the effectiveness of anti-OX40L antibody treatment in pre-sensitized mice HTX recipients [23].

By blocking the OX40–OX40L pathway, it was possible to prevent the development of cardiac allograft vasculopathy (CAV) [24]. Intragraft cellular infiltration and cytokine expression were also decreased [24]. A randomized, placebo-controlled trial involving 64 healthy volunteers evaluated the safety, tolerability, immunogenicity, and pharmacokinetic (PK) profile of the anti-OX40L antibody KY1005 (Amlitelimab) with promising results [25].

Independently, the CD28 blockade was seen as a promising approach for tolerance induction [26] and was shown to prevent allograft rejection in rodent models [27]. One of the most infamous setbacks in CD28 blockade research came from the TGN1412 trial in 2006 [28]. TGN1412 was a superagonistic CD28 antibody designed to stimulate regulatory and effector T-cells rather than block the CD28 pathway. Unlike FR104 and similar CD28 antagonists, TGN1412 acted as a potent T-cell activator, leading to a life-threatening massive cytokine storm in all six healthy volunteers during a Phase 1 trial, highlighting the need for cautious dose escalation and more rigorous preclinical assessments in humanized models [28]. Subsequently, research shifted focus toward antagonistic CD28-blocking antibodies and an indirect blockade of the CD28 pathway [29].

Cytotoxic T-lymphocyte associated protein 4 (CTLA-4), a natural immune checkpoint, has a higher affinity for CD80/CD86 than CD28, making it a potential tool for selective co-stimulation blockade. Abatacept was the first CTLA-4-Ig fusion protein (Abatacept) designed to prevent CD28-mediated co-stimulation, and it was FDA-approved in 2005 for rheumatoid arthritis (RA) and later for psoriatic arthritis and juvenile idiopathic arthritis; however, while effective for autoimmune diseases, abatacept was not potent enough to prevent rejection after organ transplantation. A humanized version of CTLA-4-Ig, which incorporates a modified IgG Fc domain, LEA29Y (Belatacept) was engineered by introducing two amino acid substitutions (L104E and A29Y) in the CTLA-4 domain, increasing its binding affinity for CD80/CD86 by 4–5 times compared to Abatacept.

Currently, Belatacept is the only FDA-approved co-stimulation blocker, and although approval by both the FDA and EMA, it is limited to kidney transplantation. The BENEFIT trial showed superior renal function and similar graft and patient survival with Belatacept-based immunotherapy compared to Cyclosporine-based immunotherapy, but a higher rate of acute rejection periods was observed with Belatacept [30]. Additionally, Belatacept was shown to increase the rate of post-transplant lymphoproliferative disease (PTLD), especially at a high dosage [31].

However, Belatacept was also investigated in heart transplant recipients, where it was associated with a low rejection rate and preserved graft function. Still, treatment was discontinued in 19% of patients because of severe side effects [32]. At the moment, two clinical trials are recruiting patients after HTX that are at risk for kidney failure (NCT04180085, NCT04477629). In a retrospective analysis, patients receiving Belatacept with reduced doses of calcineurin inhibitors (CNI) showed promising outcomes, with low rejection rates and preserved graft function [32]. In these cases, Belatacept was primarily indicated for patients with high pre-transplant donor-specific antibodies (DSAs) and those requiring renal protection. Nevertheless, approximately one-third of patients required the temporary discontinuation of Belatacept due to severe adverse effects. For highly sensitized heart transplant recipients, Belatacept still presents a potentially effective treatment option [33]. It will be necessary to await the results of these studies and to conduct follow-up studies, preclinical and clinical, to see the efficacy and safety of Belatacept in heart transplantation.

In a recent experimental setting, combining CTLA4-Ig with Notch-1 inhibition demonstrated prolonged allograft survival, while preserving immune reactivity against third-party antigens. Notably, CTLA4-Ig can have a detrimental impact on regulatory T-cell (Treg) populations by blocking CTLA-4/B7 interactions, potentially contributing to the observed increase in severe acute rejection. Notch-1 inhibition, however, is thought to stabilize Treg populations by inhibiting mTOR signaling and upregulating CTLA-4 expression, potentially preserving Treg function when used alongside CTLA4-Ig [34]. Also, combining CTLA4-Ig with an antibody that blocks TIGIT, a CD28 family coinhibitory receptor, prolongs skin allograft survival and increases the number of graft infiltrating Tregs [35]. The effect was Treg-dependent and needs to be verified in clinical SOT, especially in the heart.

## 3. Regulatory (T) Cell-Based Therapies

Alloreactive T-cells are one of the main drivers of allograft rejection and recognize donor antigens through direct, indirect, or semi-direct pathways [36]. On the contrary, there are several subsets of regulatory cells, like B-cells and macrophages, that can modulate and downregulate the immune response, and Tregs seem to be the most potent ones where clinical translation seems to be most feasible [37].

Forkhead box protein 3 (FoxP3)+ CD4+ regulatory T-cells (Tregs) are crucial for maintaining peripheral self-tolerance [38] and essential for regulating autoreactive T-cell clones. The transcription factor FoxP3 is the main regulator of Tregs, and mutations in this gene lead to often fatal autoimmune diseases in both mice and humans [39]. However, under normal conditions, Tregs are recognized for their crucial role in mediating self-tolerance, which they exert through several mechanisms in both cell contact-dependent and cell contact-independent manners [40]. First, Tregs stimulate the production of anti-inflammatory cytokines, like IL-10 and IL-35, which are associated with reduced T-cell activation. Moreover, exosomes with microRNA (miRNA) released from Tregs inhibit T effector cell proliferation and the secretion of inflammatory cytokines. Additionally, Tregs express CTLA4, which interacts with its ligand CD80/86 on antigen-presenting cells (APCs). This interaction mediates an inhibitory stimulus for T-cells. Tregs also possess the ability to directly induce apoptosis through the action of granzyme A/B and perforin and the Fas/Fas-ligand pathway. Apparently, Tregs modulate the immune response through a number of pathways.

Previous studies have demonstrated that regulatory T-cells (Tregs) are capable of inducing transplantation tolerance in murine models [7,41].

However, despite extensive research, important questions remain unanswered; for example, what the best source of Tregs is, whether to use donor- or recipient-derived Tregs, and if the Tregs need to be antigen-specific (Figure 2) [42]. An important factor is the different stability between different Treg subsets due to epigenetic changes that effect the transcription factor FoxP3, which also limits their function after in vivo transfer [43].

Ezzelarab et al. explored the fate of in vitro expanded infused Tregs and how this effects their therapeutic capacity [44]. They concluded that, while donor antigen alloreactive Tregs suppressed T-cell function in vitro, they partly lost their immunosuppressive and antiapoptotic function after being infused into heart-allografted cynomolgus monkeys.

Antigen-specific Tregs can be produced when co-culturing with donor cells/APCs pulsed with donor antigen or by transducing normal Tregs with a donor-specific transgenic T-cell receptor [41,45]. It was shown that antigen-specific Tregs are more efficient than polyclonal Tregs in reducing immune responses towards allo-antigens [45]. Alternatively, it is possible to introduce donor-specific transgenic TCRs on regular Tregs. These TCR-transduced Tregs could induce the long-term survival of heart allografts when temporarily combined with immunosuppression [41].

However, its application is constrained by the potential for mismatched hybridization between exogenous and endogenous chains [46]. A more elegant approach is using engineered Tregs that express chimeric antigen receptors (CARs) [47]. These CARs consist of a hybrid T-cell receptor (TCR), where the extracellular domain is replaced with a single-chain variable fragment derived from a B-cell receptor (BCR) with specificity to a certain target antigen [42].

In contrast to antigen-specific Tregs, CAR Tregs can be produced quickly in large numbers. Their effectiveness for inducing graft tolerance was demonstrated in several preclinical trials, with CAR Tregs directed against human leukocyte antigen (HLA)-A2 being the most promising, since mismatched HLA is only present on the allograft [48]. While first-generation CARs only had a TCR CD3j endodomain, and second-generation CARs consisted of a CD28 signaling domain that overcomes the proliferation blockade of the first generation, third generation CARs have at least two co-stimulatory molecules additional to CD28, like CD137 and OX-40 [49]. In graft versus host disease (GvHD), HLA-A2-specific CAR Tregs suppressive function on Teff proliferation and could prevent xeno-GvHD in humanized mice [50]. In a skin graft model, CAR Tregs could significantly delay rejection [51]. One major drawback of CAR Tregs is off- and on-target toxicity [49]. A study, however, showed that adding Rapamycin to the culture can reduce on-target toxicity [52]. Moreover, the Universal CAR (UniCAR) technology could possibly prevent off-target effects [53]. Here, the engineered UniCAR T-cells recognize an epitope that is not expressed under physiological conditions in circulating cells. Target molecules that express this specific UniCAR epitope can be administered into the patient. When recognizing the epitope, the CAR Tregs switch from silent mode to activation mode and are directed towards the particular antigens. At any given time, the effect of the Tregs can be inhibited by stopping administering the molecule [54]. In order to achieve clinical translation, it will be essential to test CAR Tregs in non-human primate models. A team from the University of Pennsylvania proposed a model on how to generate non-human primate CAR Tregs using artificial APCs, antigen-specific restimulation, and simian tropic lentiviral vectors [55]. It will be crucial to see the first results from using this approach. For kidney transplantation, the first clinical trial (phase I/IIa study) investigating the potency of HLA-A2-specific CAR Tregs for the prevention of rejection has already started (2019-001730-34).

While Treg cell-based therapy has shown positive effects in type 1 diabetes (T1D) and GvHD, only one clinical trial could show effectiveness in SOT [56,57]. The ONE trial showed the safety and feasibility of cellular immunotherapy in kidney transplantation [58]. The ongoing TWO study will further investigate the efficacy of Treg cell therapy [59]. Nevertheless, in heart transplantation, data for Treg cell-based therapy are scarce.

Unfortunately, most immunosuppressive drugs used after organ transplantation have negative effects on Treg function and proliferation. Calcineurin inhibitors (CNIs) reduce the numbers of circulating Tregs in liver and kidney transplant recipients [60,61]. However, combining CNIs with IL-2 therapy can salvage the numbers of Tregs and act synergistically to prevent graft rejection [62]. The expansion of Tregs also improves the efficacy of CTLA4Ig treatment and can together increase the survival of heart allografts, as already discussed earlier [63].

It is important to discover alternative immunosuppressive drugs that do not have deleterious effects on Tregs and make attempts to stimulate Treg function and numbers [64].

Adoptive Treg cell therapy has been extensively investigated. This approach involves isolating Tregs from peripheral blood, thymus, or umbilical cord, expanding them ex vivo, and subsequently, infusing them into transplant recipients. However, several challenges limit the clinical application of this technique. First, the availability of Tregs is limited, making it difficult to obtain sufficient cell numbers for therapeutic purposes. Additionally, the isolation and expansion of Tregs are technically demanding and resource-intensive processes. Alternatively, Tregs can be expanded in vivo. Injecting low-dose IL-2 can increase Treg numbers in mice and humans [65]. Surprisingly, a higher incidence of T-cell-mediated rejection was seen in patients treated with IL-2 [66]. Most likely, this is because IL-2 also increases the number of CD8+ and NK cells. This, and also the short half-life of IL-2, can be circumvented when complexing IL-2 with different monoclonal anti-IL-2 antibodies. Anti-IL-2 mAb JES6-1 can selectively increase the numbers of Tregs in vivo and was shown to prevent the rejection of islet allografts without immunosuppression [67]. In a skin transplant model, IL-2/JES6-1 complexes synergized with anti-IL-6 and Rapamycin to significantly prolong allograft survival without long-term immunosuppression [68]. Recently, Efe et al. produced a humanized IL-2 mutein that effectively prolongs skin graft survival in mice and non-human primates (NHPs) [69,70]. It can be expected that these results can also be translated to heart transplantation. Ongoing clinical trials are evaluating the safety and efficacy of Efavaleukin Alfa, a human IL-2 mutein Fc fusion protein designed to selectively stimulate Tregs in patients with GvHD (NCT03422627) [42].

## 4. Mixed Chimerism

Mixed chimerism refers to a state in which both, recipient and donor hematopoietic cells coexist in the recipient’s immune system following a bone marrow or hematopoietic stem cell (HSC) transplant following a conditioning regimen to facilitate donor cell engraftment (e.g., low-dose radiation [71], pharmacological myelosuppression [72], mega doses of BM [73], Treg treatment [74].) The goal is to induce donor-specific tolerance by allowing the recipient’s immune system to recognize the donor organ as “self”, eliminating the need for long-term immunosuppression.

This approach is based on the principle of central tolerance; when developing T-cells in the thymus are exposed to both self and donor antigens, they learn not to attack donor tissues, leading to long-lasting immune tolerance. Once donor HSCs engraft in the recipient’s bone marrow, they produce immune cells that coexist with the recipient’s own immune system. Over time, this leads to the deletion or functional inactivation of donor-reactive T-cells, preventing rejection, while maintaining immune function against infections and other antigens.

The mixed chimerism is the only tolerance approach that was successfully translated into the clinical setting by different centers [75].

Since pre-existing mature T-cells present a significant obstacle, appropriate recipient conditioning is important [5]. Full chimeras are not desired because of the risk of GVHD and because of their impacted immune system, unless the protocol is also treating, e.g., myeloma [76].

Mixed chimerism has several advantages compared to full chimerism, like lower risk of GVHD and better preservation of immunocompetence [77].

In a minor antigen mismatch porcine HTX model, mixed hematopoietic chimerism could induce long-term tolerance without immunosuppression. Swines were conditioned using a T-cell-depleting CD3 immunotoxin, thereby preventing the use of myeloablative-conditioning regimens [78]. In MHC-mismatched NHP, mixed chimerism prolonged heart allograft survival but could not achieve full tolerance [79]. Interestingly, Tregs are pivotal for maintaining tolerance of murine skin and heart grafts in a mixed chimerism model. The authors hypothesized that there might be two different tolerance mechanisms present in mixed chimerism; one that is resistant to Foxp3(+) depletion, and one that relies on Foxp3(+) cells [80].

In an NHP lung transplant model, a mixed-chimerism protocol with thymoglobulin and Belatacept could not induce donor-specific tolerance, in contrast to the successful application in renal transplantation [81]. Recently, it was shown that BM and kidney co-transplantation from the same donor could induce tolerance in NHP heart transplant recipients [82]. Interestingly, tolerant hearts revealed similar gene expression as chronically immunosuppressed recipients; although, immunosuppression was discontinued after a short time, and tolerant hearts were fundamentally different from naïve and rejected hearts [82].

In a preclinical rat model, the co-transplantation of skin and heart grafts and vascularized hind limbs could induce donor-specific tolerance without continuous immunosuppression [83]. While chimerism was diminished after 7 weeks, the animals still showed donor-specific tolerance. Solid organ co-transplantation is a promising approach for inducing tolerance and is extensively discussed below. Another interesting approach seems to be the combination of post-transplant cyclophosphamide together with sirolimus and CTLA-4 Ig. While high doses of cyclophosphamide are known to be cardiotoxic and associated with fungal and bacterial infection, there is a concentration that, combined with other drugs, can prevent GVHD and cause alloreactive T-cells to be functionally impaired [84,85,86].

Kabore et al. could show that low-dose cyclophosphamide induces mixed chimerism and graft acceptance in a mismatched murine bone marrow transplantation model, when combined with sirolimus and CTLA-4 Ig or lymphocyte depletion [87]. It will be of great interest to see whether this approach could also show effectiveness in heart transplantation.

A clinical study investigated the effect of concomitant donor bone marrow infusion and heart transplantation and showed a reduced rate of acute rejection and cardiac allograft vasculopathy [88]. However, this effect was seen after steroids were weaned, but immunosuppression could not be discontinued completely. Importantly, mixed chimerism was only minimally detectable at 3 months post-transplantation, suggesting that mixed chimerism may be feasible in heart transplantation, but bone marrow transplantation (BMT) is insufficient in promoting long-lasting donor-specific tolerance alone.

The mixed chimerism approach offers a potential cure for transplant rejection, allowing for donor-specific tolerance without lifelong immunosuppression. While successful in kidney transplantation, its application in heart transplantation remains challenging due to the complexity of bone marrow transplantation. Ongoing research aims to refine conditioning protocols and immunomodulatory strategies to make this a viable option for broader clinical use.

## 5. Thymus Transplantation

A promising strategy in transplant immunology involves cultured thymus tissue implantation (CTTI), which addresses challenges in immune tolerance and thymic insufficiency [89]. This method entails the ex vivo preparation of thymic tissue from the donor, where thymus slices are cultured to deplete mature donor T-cells. This depletion minimizes the risk of GVHD and provides a conducive environment for the development and education of recipient-derived T-cells in the implanted thymic tissue [90].

Following the culturing phase, the processed thymic tissue is surgically implanted into the recipient, typically within a muscular site such as the quadriceps. This localized implantation ensures a supportive microenvironment for the engraftment and function of the thymic tissue. The approach has demonstrated clinical efficacy, notably in patients with DiGeorge syndrome (22q11.2 deletion syndrome), a condition characterized by thymic aplasia and profound immunodeficiency [89,91].

In experimental models, such as rats, the integration of CTTI into transplantation protocols has shown remarkable potential. Combining CTTI with heart transplantation and thymectomy successfully induced donor-specific tolerance in rats without immunosuppression [92]. While neither the CTTI group nor the control group rejected the allografts, because all animals were T-cell depleted prior to transplantation, only the CTTI-group rejected third-party grafts, indicating donor specificity in the CTTI-group. Notably, this was achieved without the need for continued immunosuppression. These findings underscore the role of CTTI in reconstituting a functional thymic environment capable of producing tolerant, self-restricted T-cells, thereby promoting long-term graft survival, while reducing dependency on immunosuppressive therapies. In 2022, the first combined heart transplantation and cultured thymus tissue implantation was performed in a child [93].

Another approach is to perform intrathymic injections with donor splenocytes before transplantation. A group performed intrathymic pretreatment with allogeneic spleen cells and simultaneous T-cell depletion by antilymphocyte serum and detected durable donor-specific tolerance [94]. However, similar studies, for example with intrathymic injection of the anti-Fas monoclonal antibody, could not observe donor-specific tolerance, especially across MHC barriers [95,96]. An alternative approach that has garnered attention is the concept of “thymoheart” transplantation [97]. In this method, minced donor thymic tissue is implanted into the recipient’s cardiac adipose tissue and the right atrial appendage prior to transplantation. This strategy has demonstrated the potential to induce donor-specific tolerance in pig models. However, its clinical applicability remains limited due to the requirement for donor preparation 60 to 90 days before transplantation, presenting significant logistical challenges. Similarly, the strategy of thymus/heart co-transplantation has not progressed to clinical application yet, primarily due to its inability to achieve tolerance. In a preclinical study involving swines, all transplanted heart grafts were ultimately rejected despite this approach [98]. In contrast, en bloc heart/thymus transplantation in pigs demonstrated prolonged graft survival up to the study endpoint without the need for continuous immunosuppression [99]. However, donor-specific tolerance was not proven in these studies. Similarly, attempts at en bloc transplantation of the thymus, heart, and chest wall in mice failed to induce donor-specific tolerance, rendering this approach largely unsuccessful [100].

## 6. Double Organ Transplantation

Double organ transplantation (also known as combined organ transplantation) involves transplanting two different organs from the same donor into a single recipient. This approach has gained attention not only for addressing multi-organ failure but also as a potential strategy for immune tolerance induction in solid organ transplantation.

While it may seem contradictory at first, combined heart–kidney transplantation has been shown to reduce the rates of cellular rejection and recipients of double organ transplantation were more likely to have freedom from cardiac allograft rejection and have acceptable survival rates [101,102]. Some organs, particularly the liver, have inherent immunomodulatory properties that can promote donor-specific tolerance when transplanted alongside another organ. The idea is that the immunosuppressive environment created by one organ (e.g., the liver) may extend tolerance to the other organ (e.g., the heart, kidney, or lung), reducing the risk of chronic rejection and the need for lifelong immunosuppression. Kidneys have a much higher density of MHC class II and I antigens compared to hearts [103]. There is reason to suspect that the presence of an organ with a higher density of comparable antigenic markers may facilitate partial tolerance toward an organ exhibiting lower antigenic levels. A study suggested that both an increase in donor antigen load and an effect specific to the kidney allograft contribute to the positive outcomes seen in pigs receiving kidney and heart co-transplantation [104]. In a mechanistic study, the authors could show that kidney-induced cardiac allograft tolerance (KICAT) is highly dependent on Foxp3+ T-cells, but that graft acceptance is independent from the thymus [105]. They could induce prolonged survival of co-transplanted cardiac allografts without immunosuppression. However, tolerance was not achieved in skin grafts, showing the different immunogenicity of the skin. Interestingly, CAV was not prevented by this protocol, raising the question of the exact pathomechanisms of CAV in allograft rejection.

Looking at the liver, it seems like the hepatic allograft can also positively affect rejection episodes by promoting the acceptance and immunotolerance of other allografts [106,107]. Prior liver transplantation was associated with a reduction in allograft-mediated immune injury and less CAV in subsequent heart transplantation [108,109].

The mechanisms behind this are not fully understood, yet the number of soluble HLAs in peripheral blood, donor myeloid cells that can potentially induce donor leukocyte micro chimerism, and enhanced Treg activation are some of the potential mechanisms [110].

## 7. Innovative Advancements for the Extension of the Donor Pool

### AB0-Incompatible Transplantation

ABO-incompatible (ABOi) heart transplantation has been explored to address scarcity of suitable donor organs in children. Infants under 12 months of age have an immature immune system and do not produce significant levels of anti-A or anti-B antibodies, allowing them to accept hearts from donors with different blood types without a higher risk of rejection. Studies have demonstrated that survival rates and freedom from rejection in ABOi heart transplants are comparable to those in ABO-compatible (ABOc) transplants [111,112].

In adults, ABOi heart transplantation remains challenging due to the mature immune system’s production of anti-A and anti-B antibodies, which can lead to hyperacute rejection. Most successful cases in adults have been inadvertent ABOi transplants with low antibody levels, but these are associated with early rejection and poor outcomes [113,114].

By now it is possible to, with the use of specific enzymes, convert the antigens of blood group A from the human renal vasculature to O-type. MacMilan et al. used a hypothermic and normothermic perfusion machine model to achieve a reduction of blood group A antigen by about 80% within two hours by using FpGalNAc deacetylase and FpGalactosaminidase [115]. Using these enzymes revealed no signs of vascular damage, meaning that FpGalNAc DeAc and FpGalNase were safe in this setting.

When testing an ex vivo model of AB0-incompatible reperfusion, the group showed that there was no complement activation in the AB0-incompatible enzyme-treated kidneys. There was less C4d and C5b-9 deposition on the microvasculature and that these kidneys did not bind circulating anti-A antibodies, in contrast to AB0-incompatible controls. Interestingly, the conversion of antigens in enzyme-treated AB0-incompatible kidneys was higher in lungs than in kidneys, where a 97% conversion rate was achieved [116]. MacMilan et al. speculated that this was due to a different method of fluorescence quantification of fixed tissue [115]. Still, using an ex vivo lung perfusion system, similar results could be achieved with less complement deposition, antibody binding, and antibody-mediated injury compared with control lungs [116].

A review looked at AB0-incompatible living donor liver transplantation (LDLT) in 11 centers within a timespan between 2010 and 2016 [117]. While there were no new technologies, treatment protocols were developed further. Rituximab administration prior to AB0-incompatible living donor liver transplantation and plasma treatment procedures have proven to be advantageous [117]. A promising approach is the dual split LDLT, where AB0i and AB0c grafts are used simultaneously [118].

Additionally, the concept of intraoperative immunoadsorption is very appealing, as it reduces the amount of allogeneic blood and blood products that patients are exposed to compared to the plasma exchange technique and, thus, reduces transfusion-related morbidity [119]. Intraoperative immunoadsorption involves circulating the patient’s blood through an adsorption column during cardiopulmonary bypass, where antibodies are selectively removed from the plasma. This targeted antibody depletion reduces the risk of hyperacute rejection, while preserving other plasma components, and certainly has great potential to benefit pediatric heart transplant recipients.

While AB0-incompatible transplantation, especially in the heart, is still in its early beginnings, it will inevitably contribute to a larger donor pool for SOT in the future.

## 8. Discussion and Future Perspectives

The pursuit of tolerogenic therapies in cardiac transplantation remains an essential yet challenging objective due to the highly immunogenic nature of the heart and the limitations of current immunosuppressive regimens. While co-stimulatory blockade, mixed chimerism, and regulatory T-cell therapies have shown promise in preclinical and early clinical studies, their translation into routine clinical practice requires further optimization.

Despite the progress made in understanding immune tolerance, heart transplantation lags behind kidney and liver transplantation in the successful clinical application of tolerogenic strategies. The inherent immunogenicity of the heart, along with its susceptibility to antibody-mediated rejection and chronic allograft vasculopathy, presents unique barriers that must be overcome [8]. Advances in our understanding of alloimmune responses, particularly memory T-cell responses and humoral rejection, highlight the need for targeted approaches that go beyond standard immunosuppressive regimens.

A critical avenue for future research is refining combination therapies that integrate co-stimulation blockade, Treg-based therapies, and mixed chimerism. Emerging data suggest that co-stimulation blockade, particularly with newer-generation anti-CD40L agents, could be key to preventing acute rejection, while preserving immune tolerance [16]. However, long-term studies assessing their efficacy in preventing chronic rejection and CAV are necessary. The role of regulatory T-cells, both in endogenous expansion and ex vivo engineered approaches, such as CAR-Tregs, is another promising area that warrants further exploration. Additionally, novel approaches, such as IL-2 muteins, which selectively expand Tregs, while minimizing effector T-cell activation, hold potential for long-term graft survival with minimal immunosuppression [70].

Mixed chimerism remains the only strategy to date that has successfully been translated into the clinical setting for tolerance induction in solid organ transplantation, primarily in kidney transplantation [75]. However, achieving stable chimerism in heart transplantation has been met with difficulties, particularly due to the need for intensive conditioning regimens that pose significant risks to the recipient. Recent studies investigating post-transplant cyclophosphamide and sirolimus in combination with donor hematopoietic stem cell infusion offer promising alternatives for achieving transient or sustained mixed chimerism with reduced toxicity [87]. Further research into optimizing these regimens for cardiac transplantation is required.

Another critical frontier in cardiac transplantation is the expansion of the donor pool. Innovations in xenotransplantation, particularly with genetically modified pig hearts, have demonstrated feasibility in recent experimental models [120]. While significant immunological barriers remain, advancements in gene editing and immunomodulatory strategies may enable broader clinical application in the future [121]. Similarly, ABO-incompatible transplantation, long considered a feasible approach in pediatric recipients due to their immature immune systems, may become more widely applicable in adults with the advent of enzymatic antigen modification and enhanced desensitization protocols [114].

Beyond the holy grail of immunological tolerance, addressing the long-term complications of cardiac transplantation is imperative. The high incidence of post-transplant malignancies, infections, and metabolic disorders associated with lifelong immunosuppression underscores the need for safer, more targeted therapeutic strategies. Advances in biomarker development and immune profiling could enable personalized immunosuppression regimens, minimizing toxicity, while maintaining graft function.

Ultimately, the future of cardiac transplantation will be shaped by interdisciplinary research efforts combining immunology, bioengineering, and regenerative medicine. The integration of cellular therapies, gene-editing technologies, and precision immunosuppression strategies will be crucial in achieving true donor-specific tolerance. While significant hurdles remain, continued innovation and collaboration within the field offer hope for a future where heart transplant recipients can achieve long-term graft survival without the burden of lifelong immunosuppression.

## Figures and Tables

**Figure 1 ijms-26-03968-f001:**
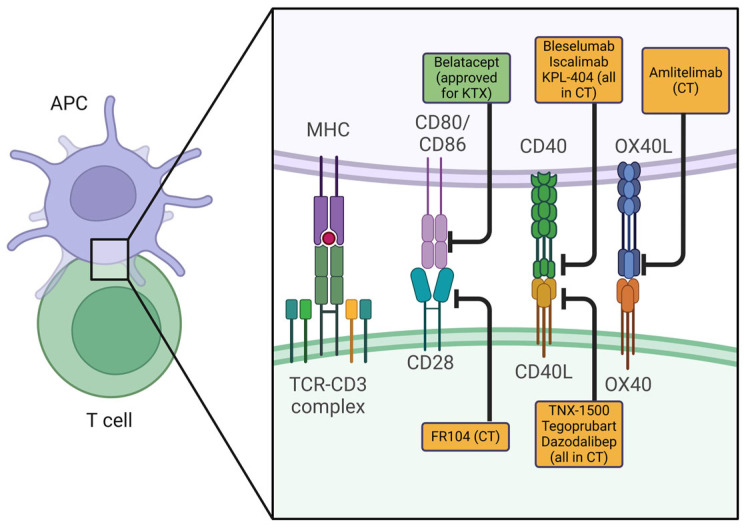
Selected biologicals blocking the co-stimulatory molecules on T-cells and antigen-presenting cells. The green box indicates approved co-stimulation blockers; yellow boxes indicate co-stimulation blockers in development. Abbreviations: APC: antigen-presenting cell, KTX: kidney transplantation, CT: clinical trial. Created in BioRender. Pilat-michalek, N. [2025] https://BioRender.com/c33c247.

**Figure 2 ijms-26-03968-f002:**
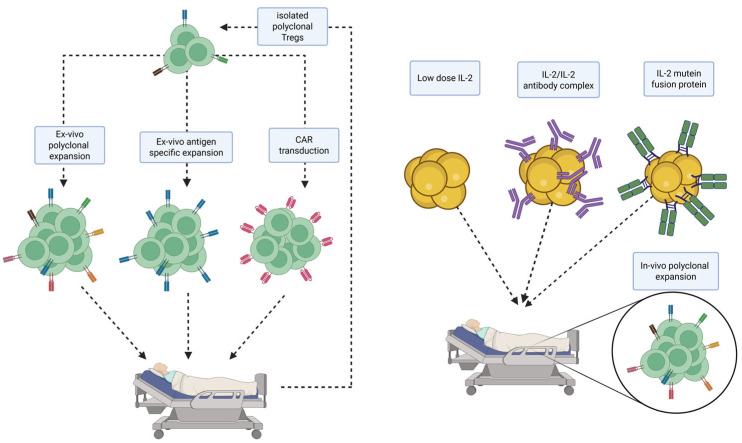
Selection of therapeutic approaches for Treg therapy. Created in BioRender. Pilat-michalek, N. [2025] https://BioRender.com/mg7o6kd.

**Table 1 ijms-26-03968-t001:** Selection of clinical trials investigating tolerance approaches in organ transplantation (clinicaltrials.gov.org accessed on 3 March 2025).

Trial	Trial Name	ClinicalTrials.gov ID	Phase	Condition	Status	Key Findings
Belatacept	Belatacept Evaluation of Nephroprotection and Efficacy as First-Line Immunosuppression (BENEFIT)	NCT00256750	Phase III	Kidney transplantation	Completed	Better renal function than Cyclosporine but with higher acute rejection rates.
KY1005 (Amlitelimab)	A Study of Subcutaneous KY1005 in Healthy Volunteers	NCT04449939	Phase I	Healthy volunteers	Completed	Safe, well-tolerated, and showed promising pharmacokinetics.
TNX-1500 (Anti-CD40L mAb)	TNX-1500 (Fc-modified Humanized Anti-CD40L mAb) Single Ascending Dose Study in Healthy Participants	Not available (pending NCT)	Phase I	Healthy volunteers (kidney transplantation dosing)	Completed (Feb 2025)	Blocked antibody responses, no thromboembolic events, and supports monthly dosing. Phase 2 planned.
Tegoprubart	Long-Term Safety and Efficacy of Tegoprubart in Kidney Transplant Recipients	NCT06126380	Phase II	Kidney transplantation	Ongoing	Non-thrombogenic, Phase II ongoing for efficacy assessment.
Dazodalibep + Belatacept	A Study to Evaluate the Safety and Efficacy of Dual Co-Stimulation Blockade With VIB4920 and Belatacept for Prophylaxis of Allograft Rejection in Adults Receiving a Kidney Transplant	NCT04046549	Phase II	Kidney transplantation	Ongoing	Dual blockade approach under Phase II evaluation.
Belatacept	Belatacept in Heart Transplant Recipients at Risk for Kidney Failure	NCT04180085	Recruiting	Heart transplantation + kidney failure risk	Recruiting	Assessing Belatacept’s efficacy in heart transplant patients with kidney risk.
Belatacept	Belatacept in Heart Transplant Recipients at Risk for Kidney Failure (Second Cohort)	NCT04477629	Recruiting	Heart transplantation + kidney failure risk	Recruiting	Ongoing trial similar to NCT04180085, focusing on safety and efficacy.

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
