# Peer review of "Tolerogenic Therapies in Cardiac Transplantation"

_ijms, 2025, doi:10.3390/ijms26093968_

Round 1

Reviewer 1 Report

Comments and Suggestions for Authors

A very good paper. Not much work for a reviewer to do...

  • The immunoadsorption method for ABO incompatible HTX deserves mentioning,
  •  The authors may find one of the practical refs below useful.

1: Issitt RW, Cudworth E, Cortina-Borja M, Gupta A, Kallon D, Crook R, Shaw M,
Robertson A, Tsang VT, Henwood S, Muthurangu V, Sebire NJ, Burch M, Fenton M.
Rapid desensitization through immunoadsorption during cardiopulmonary bypass. A
novel method to facilitate human leukocyte antigen incompatible heart
transplantation. Perfusion. 2024 Apr;39(3):543-554. doi:
10.1177/02676591221151035. Epub 2023 Jan 10. PMID: 36625378; PMCID: PMC10943618.

2: Issitt R, Crook R, Shaw M, Robertson A. The Great Ormond Street Hospital
immunoadsorption method for ABO-incompatible heart transplantation: a practical
technique. Perfusion. 2021 Jan;36(1):34-37. doi: 10.1177/0267659120926895. Epub
2020 Jun 3. PMID: 32493108; PMCID: PMC7770210.

3: Robertson A, Issitt R, Crook R, Gustafsson K, Eddaoudi A, Tsang V, Burch M. A
novel method for ABO-incompatible heart transplantation. J Heart Lung
Transplant. 2018 Apr;37(4):451-457. doi: 10.1016/j.healun.2017.05.006. Epub 2017
May 5. PMID: 28554587.

---------------------------------------

Further comments from the Journal's guided review questions

• Do you consider the topic original or relevant to the field? Does it
address a specific gap in the field? Please also explain why this is/ is not
the case. Yes, adequate topic that addresses a gap. 
• What does it add to the subject area compared with other published
material? Good topic for a review, not too big not too small. I have not seen many papers like this recently so it is nice to see all these thigs in one place, all coming from a very experienced clinical group. 
• Are the conclusions consistent with the evidence and arguments presented
and do they address the main question posed? Please also explain why this
is/is not the case. This question is not very apposite since the paper is a review, not original research - so not much of a 'conclusion based on results'.
• Any specific improvements should the authors consider discussing? Not really, from my point of view. 
• Any additional comments on the tables and figures? No. 

Reviewer 2 Report

Comments and Suggestions for Authors

In this review, the authors discuss the recent status of tolerogenic therapies with a focus on heart transplantations in humans. This field has advanced a lot in the last decades even as less heart transplantation are performed in recent years.

I still have some comments:

  • In the abstract I am missing a bit a attenuated statement that more and more medical treatments are available for heart pathologies rendering heart transplantation a less-often needed treatment (and the patients in need for it, might be way more ill).
  • The figures are nice an informative, but their resolution need to be improved
  • Some abbreviations are not introduced (e.g. CAV)

Reviewer 3 Report

Comments and Suggestions for Authors

In the review “Tolerogenic Therapies in Cardiac transplantation” by Wolner at al, authors analyze and report the existing and novel approaches to achieve tolerance. This is a well-designed and substantial work with a great potential interest for readers. However, manuscript would benefit from some minor edits:

1 Ref 39 is not great. It is very outdated (8 years old), moreover, the review has some controversial and not so well supported statements. Relatively fresh and good review by Sakaguchi such as PMID: 37400628 (or similar) would be better to use there.

2 It may be worth to mention another review by Ford (PMID: 35902775) or, at least, to mention the existence of direct, indirect and semi-direct allorecognition mechanisms, or their critique. This review also has an important point of why the majority of successful experimental transplant models fail in clinical application.

3 Figure 2, it is not very clear from the graphic why one recipient lays on the bed and another one stands. It looks like the main difference between two sets of therapeutic approaches is to amplify recipient’s Treg in vivo or create them in vitro, but the corresponding pictures are not helpful to see that difference.

4 Line 214, ref 42. Please clarify that there were in vitro expanded Tregs.

5 In Treg -based therapies, it may be worth to mention these (or similar) reviews: PMID 37397971, 37153574, and discuss, at least shortly, the problem of Treg stability/durability.
